# A Comprehensive Clinical Outcome Analysis of Endoscopic Vessel Harvesting for Coronary Artery Bypass Surgery

**DOI:** 10.3390/jcm13123405

**Published:** 2024-06-11

**Authors:** Hari Kumar Sampath, Terence Ji Hui Lee, Chua E. Cher, Shen Liang, Ooi Oon Cheong, Theo Kofidis, Sorokin Vitaly, Faizus Sazzad

**Affiliations:** 1Department of Surgery, Yong Loo Lin School of Medicine, National University of Singapore, Singapore 119228, Singapore; 2Department of Cardiac, Thoracic and Vascular Surgery, National University Heart Centre, Singapore 119228, Singapore; 3Biostatistics Unit (BSU), Department of Medicine, National University of Singapore, Singapore 117549, Singapore; 4Centre for Translational Medicine (MD6), National University of Singapore, 14 Medical Drive, Level-8 (South), Singapore 117599, Singapore

**Keywords:** vein harvesting, endoscopic, endoscopic vein harvesting, open vein harvesting, coronary artery bypass, bypass grafting

## Abstract

**Background:** The long saphenous vein is routinely used for coronary bypass graft (CABG) surgery, and two primary techniques are commonly utilized: endoscopic vessel harvesting (EVH) and open vessel harvesting (OVH). The aim of this study was to compare the clinical outcomes of the EVH and OVH techniques used for CABG within the confines of a tertiary hospital. **Methods:** The clinical data of all patients subjected to either EVH or OVH for CABG surgery between 2014 and 2018 were retrospectively analyzed. Statistical analysis was performed to discern variations in the rates of postoperative complications between EVH and OVH. **Results:** A cohort of 1884 individuals were included in this study, 75.3% of whom underwent EVH. Notably, the incidence of postoperative leg wound complications was significantly different between the patients who underwent OVH and the patients who underwent EVH, with incidence rates of 18.6% and 32%, respectively (*p* < 0.001). Leg wound complications (*p* < 0.001; OR 1.946; 95% CI 1.528–2.477) and leg wound infections (*p* = 0.050, OR 1.517, 95% CI 0.999–2.303) were significantly associated with OVH. Moreover, leg wound hematoma (*p* = 0.039, OR = 0.402, 95% CI = 0.169–0.957) and EVH were strongly associated. **Conclusions:** The large sample of patients and the inclusion of a range of Asian ethnic groups provided notable insights into postoperative complications related to different modalities. EVH was associated with a lower incidence of postoperative leg wound complications, which suggests that EVH is a better modality for those undergoing CABG surgery.

## 1. Introduction

Coronary artery bypass grafting (CABG) is a surgical intervention that is commonly performed to treat obstructed atherosclerotic heart vessels [1]. The long saphenous vein (LSV) is a commonly employed conduit, and LSV grafts are used in more than 90% of CABG procedures [2,3]. Open vessel harvesting (OVH) is a conventional LSV harvesting method in which the leg is incised to its full length [4]. As a less invasive alternative, endoscopic vessel harvesting (EVH) has the potential to mitigate complications due to its distinct advantages. EVH is associated with a significantly lower incidence of postoperative complications such as leg wound infections and other wound complications like hematomas, seromas, and dehiscence. These reduced complications contribute to better overall patient outcomes [5]. The cosmetic benefits of EVH are noteworthy, as smaller incisions are less conspicuous, leading to a better cosmetic appearance. A shorter hospital stay is an additional benefit, potentially reducing costs and facilitating a quicker return to daily activities. Patients undergoing EVH typically experience less postoperative pain, quicker recovery times, and faster mobilization, which are crucial for reducing hospital stays and overall healthcare costs. Current guidelines from the Society of Thoracic Surgeons (STS) and the European Association for Cardio-Thoracic Surgery (EACTS) support the use of EVH due to these benefits [6,7]. EVH is thus often favored over OVH in certain clinical scenarios due to its numerous advantages. EVH is particularly advantageous for high-risk patients, such as those with diabetes, obesity, or peripheral vascular disease, as well as for elderly patients and those requiring multiple grafts. The reduced morbidity and faster recovery associated with EVH make it a preferred method in many hospitals and surgical centers, which have incorporated it into their standard practices and training programs for cardiovascular surgeons.

However, EVH necessitates specialized training and proficiency, so its steep learning curve possibly prolongs the surgery and increases the incidence of complications [8,9,10]. The use of diverse EVH devices and techniques increase the complexity of the procedure, influencing outcomes and impacting procedural efficacy [5]. There are still concerns about the possibility of blood vessel damage during EVH, as the procedure potentially jeopardizes graft quality. Moreover, as there are no direct head-to-head comparisons between EVH and OVH, there are challenges in definitively establishing the superiority of EVH across all scenarios [11,12]. Notwithstanding these limitations, ongoing research and technological advancements are likely to address these issues, enhancing the overall effectiveness and applicability of EVH in cardiovascular surgery. The aim of our study was to directly compare the clinical outcomes of EVH with those of OVH to uncover potential disparities in clinical application.

## 2. Materials and Methods

This retrospective cohort study was designed to compare the clinical outcomes of patients who underwent either endoscopic vascular harvesting or open vessel harvesting for coronary artery bypass surgery. For the purpose of this study, data were acquired from 1994 patients who underwent open heart surgery for coronary artery revascularization at a single tertiary cardiac institute between 1 January 2014 and 31 December 2018. This study was approved by the Domain Specific Review Board of the National Healthcare Group (#2019/00578), and the requirement for informed consent was waived.

Given the number of EVH procedures performed, the number of recruited patients depended on the case volume, which has increased since endoscopic harvesting techniques are more commonly performed in the center. Comparable control participants were chosen from the same patient cohort that underwent OVH procedures at the same facility. Patients who did not meet the criteria for saphenous vein utilization for CABG were excluded. Patients who underwent total arterial CABG or a single graft from the internal mammary artery were also excluded.

### 2.1. Data Collection

The Computerized Patient Support System (CPSS), established by the National University Hospital in 1998, serves as an intricate medical registry. Patients’ medical admissions, medication, and surgical histories were consolidated in the system. For this study, data retrieved from the CPSS were aggregated and seamlessly integrated with REDCAP, a secure and user-friendly web platform designed for database compilation and management.

### 2.2. Clinical Characteristics

Several baseline characteristics were collected for analysis, including age, sex, ethnicity, smoking history, hypertension status, renal disease status, hyperlipidemia status, diabetes status, peripheral vascular disease status, carotid disease status, mobility status, and operative urgency. Operative details of CABG, such as LSV harvesting techniques and the instruments used, were documented. Comprehensive data on EVH procedures, including CO_2_ insufflation amount and incision size, were gathered. Postoperative measures, such as T.E.D.^TM^ (CardinalHealth, Dublin, OH, USA) stockings and immediate postoperative care, were also recorded.

### 2.3. Vein-Harvesting Techniques

The LSV, an essential conduit used in CABG, was previously accessible only through OVH, which required an incision from the ankle to the groin. This procedure involves palpating the LSV as it passes through the medial malleolus and following its path toward the knee next to the medial tibial border. A 3–5 cm incision was made, starting proximal to the medial malleolus and ending at the location selected for the procedure and complete revascularization. Branch ligation and dissection of the long saphenous vein were then performed. The LSV is then carefully insufflated to look for any signs of leakage or scarring. After hemostasis, the leg incision can be closed using intracutaneous sutures made of finely absorbable monofilaments.

The great saphenous vein is comprehensively assessed by reviewing the patient’s medical history and conducting a physical examination before endoscopic harvesting of the LSV. If needed, a lower limb venous ultrasound scan was performed, and an on-table ultrasound assessment was performed for surgical planning. EVH involves first locating the saphenous vein either manually or via ultrasound assistance. Then, a 2.5 cm incision is made at the medial aspect of the leg, and tissues are dissected to visualize the vein. After the saphenous vein is identified, a balloon tip trocar is inserted, and CO_2_ is insufflated to facilitate tunnel creation.

A subcutaneous tunnel is created by dissecting the saphenous vein and tributaries from the surrounding fat. Careful management is crucial for avoiding inadvertent injuries, and the procedure is tailored to accommodate variations in vein anatomy (Figure 1). After the tunnel is successfully created, the saphenous vein is harvested using an endoscopic device, ensuring proper electrocautery and division of tributaries. The proximal aspect of the saphenous vein is then ligated via a separate “stab and grab” incision toward the end of the tunnel. After the saphenous vein is extracted, the proximal end is cannulated and gently insufflated to minimize endothelial damage. Any avulsions observed were then repaired with sutures. The vein is finally flushed to remove any residual clots. Finally, a compression bandage is placed around the leg after the wound is closed.

### 2.4. EVH Devices 

The Vasoview™ 7xB EVH System by Maquet, Getinge Group (Gothenburg, Sweden), and the VirtuoSaph™ Plus EVH System by Terumo Cardiovascular (Ann Arbor, MI, USA) offer advanced solutions for endoscopic vessel harvesting, each with their own distinct features and functionalities. While both systems aim to optimize endoscopic vessel harvesting, they differ in their design nuances, such as dissection techniques and device ergonomics, catering to varied surgeon preferences and clinical settings. The Vasoview^TM^ 7xB EVH system) provides high-definition endoscopic visualization and employs a blunt dissection technique, minimizing tissue trauma while also incorporating bipolar electrocautery for efficient hemostasis. Its ergonomic design enhances surgeon comfort and control, with the added benefit of CO_2_ insufflation for a clear operative field (Figure 1). The VirtuoSaph^TM^ Plus EVH System) similarly prioritizes precise visualization and utilizes a semi-blunt, atraumatic dissection approach to preserve vein integrity. It features an intuitive control interface and single-use components for sterility maintenance, along with a disposable bipolar cautery system for hemostasis.

### 2.5. Statistical Analysis

Our data analysis utilized IBM’s Statistical Package for the Social Sciences version 27.0 (SPSS v27). Our descriptive analysis covered all captured variables. Continuous variables were presented as medians and interquartile ranges (IQRs) for non-normally distributed ones and as means and standard deviations (SDs) for normally distributed ones. Frequency tables, including N and percent, were reported for categorical variables. Baseline variables were compared using appropriate tests: the Student’s *t*-test or Mann–Whitney U test for continuous variables, and the χ2 test for categorical variables. The paired *t*-test assessed continuous variables, while McNemar’s test compared categorical variables. Univariate and multivariate logistic regression analyses were used to determine the association between vessel-harvesting methods (EVH vs. OVH) and leg wound complications. By adjusting for confounders and prognostic factors, these analyses provide an accurate estimate of the harvesting method’s independent effect. This approach yields adjusted odds ratios, indicating the strength and significance of the association, ensuring reliable results. Logistic regression was used to compare operational outcomes between the two harvesting methods to adjust for important prognostic factors or confounders detected using bivariate analysis. The significance level for all analyses was set at 0.05.

## 3. Results

Of the initial 1935 patients, 1884 were analyzed for postoperative complications, with 51 excluded due to mixed or converted procedures. Moreover, 75.3% underwent EVH, while 24.7% underwent OVH. The sample included 14.7% females, with 65.9% of Chinese ethnicity.

### 3.1. Baseline Characteristics

Before analyzing postoperative complications, we assessed the preoperative baseline characteristics of our study population. Table 1 summarizes clinically relevant factors. This analysis aimed to ensure comparability between the EVH and OVH groups. Significant differences were observed in ejection fraction (EF), operative urgency, and logistic EUROscore.

### 3.2. Clinical Outcomes

In the overall clinical outcomes of 1884 CABG patients, renal complications were low, with 2.2% experiencing acute kidney injury (AKI) and 0.6% suffering renal failure. Cardiac-related deaths accounted for 0.6% (12 patients), with no perioperative myocardial infarctions reported. Saphenous vein usage prevailed (73.6%), with 4.5% utilizing the radial artery. Most patients had three distal coronary anastomoses (56.8%) and underwent a cardiopulmonary bypass (97.1%), primarily through aortic and atrial/caval cannulation. The Vasoview™ 7xB EVH system was predominantly used for EVH, with left leg incisions being the most common (64.5%). The VirtuoSaph™ Plus EVH System was used in 2.8% of procedures. Most EVH incisions were 2.5 cm, with 63.8% of vessels deemed of good quality (Figure 2). Further details can be found in Appendix A.

### 3.3. EVH vs. OVH

Table 2 presents bivariate comparisons of clinical outcomes between the study groups. EVH was associated with a significantly higher number of arterial and venous grafts, a longer procedure, cumulative bypass time, and cumulative cross-clamp times (*p* < 0.001 for all). Postoperative leg wound infections were less frequent with EVH (5.3% vs. 8.2%, *p* < 0.001), while EVH showed a higher incidence of leg wound hematoma compared to OVH (3.2% vs. 1.3%, *p* < 0.001). No significant differences were observed in other complications, renal failure, stroke, discharge status, readmission within 30 days, cardiac mortality, or major adverse cardiovascular events (MACEs).

Univariate and multivariate logistic regressions, adjusting for prognostic factors and confounders, demonstrated a significant association between leg wound complications and harvesting method. OVH was significantly associated with leg wound complications (*p* < 0.001; OR 1.946; 95% CI 1.528–2.477), including leg wound infections (*p* = 0.050; OR = 1.517; 95% CI = 0.999–2.303). Conversely, leg wound hematoma was strongly associated with EVH (*p* = 0.039; OR = 0.402; 95% CI = 0.169–0.957). Further details are available in Appendix A.

## 4. Discussion

Our investigation revealed that, compared with OVH, EVH was associated with a lower incidence of postoperative complications, particularly leg wound infections. This trend aligns with previous findings [12,13,14]. The lower incidence of such complications may be because the incision for EVH is shorter. Conducted on a broad scale, this expansive retrospective investigation was designed to delve into the postoperative ramifications encountered by individuals who underwent either EVH or OVH for CABG surgery. Notably, EVH has emerged as the prevailing standard of care for patients necessitating saphenous vein grafts for coronary bypass procedures, a recommendation put forward by the International Society of Minimally Invasive Cardiothoracic Surgery [15]. This recommendation is the result of studies proving the positive association between EVH and the lower incidence of leg wound complications, and there is no notable difference in the long-term incidence of significant adverse cardiac outcomes when compared with the outcomes associated with OVH. In a meta-analysis, Deppe et al. also reported that EVH is safe for patients undergoing CABG surgery as it reduces leg wound infections. They analyzed a total of 27,789 patients from 43 studies (16 RCTs and 27 observational trials) who underwent saphenectomy: 46% EVH and 54% OVH [16].

While previously reported rates of hematoma occurrence for EVH and OVH are comparable [17,18,19], our comprehensive investigation revealed a statistically significant difference in hematoma incidence between the two methods. In our study, the frequency of hematoma development was higher in the EVH group than in the OVH group. Several factors could contribute to this disparity. Inadequate lighting and visualization might impede the precise dissection of saphenous vein tributaries during EVH, potentially leading to hematoma formation [6,20]. Additionally, the learning curve associated with EVH, particularly with devices such as Vasoview^TM^ (Marquet) and VirtuoSaph^TM^ (Terumo), could prolong surgeon adaptation, causing more reluctance to perform the operation. Hematoma may be more prevalent during this acclimation period, underscoring the complex interplay of factors affecting postprocedural outcomes [21]. Furthermore, a large retrospective multicenter analysis has indicated that EVH has a lower risk of mortality compared to OVH over a follow-up period of six years. Although our study did not include an extended follow-up period, similar findings are expected from this cohort as reported in other studies [22].

In our study, the incidence of postoperative pain or edema was not significantly lower in the EVH group than in the OVH group. This finding differs from that in several studies reporting a notably lower incidence of postoperative pain [23,24,25,26]. This discrepancy could be attributed to the idea that pain may not solely stem from differences in incision length between different harvesting methods but rather from injury to adjacent tissues—a factor common to both EVH and OVH. Furthermore, edema was observed during both harvesting methods, likely because the removal of saphenous veins does not involve the deep venous drainage system.

### Limitations

This study has a few limitations. These include potential biases due to its retrospective nature and single-center design, which precluded randomization or blinding, potentially introducing bias between the two groups. The absence of randomization might have contributed to baseline disparities that could influence the study outcomes. Nevertheless, there were no substantial differences in baseline characteristics between the EVH and OVH groups, except in cases of operative urgency. However, the groups did differ in significant clinical indicators such as logistic EUROscore, ejection fraction, and left main stem disease, which may have affected the clinical results. Notably, patients requiring urgent operations preferred OVH over EVH, likely because OVH is a faster procedure. Additionally, since the investigation was conducted in a single center, the patient population might not be representative of the general population.

## 5. Conclusions

Our findings suggest that EVH reduces the incidence of postoperative complications, providing benefits such as a lower risk of complications, improved cosmetic outcomes, and faster recovery in CABG surgery. However, the consideration of operative urgency and the necessity for specialized training in EVH should inform its implementation. To solidify these findings, further prospective studies are crucial. Future research should focus on long-term outcomes and cost-effectiveness and identify the optimal conditions for EVH to enhance care quality for CABG patients.

## Figures and Tables

**Figure 1 jcm-13-03405-f001:**
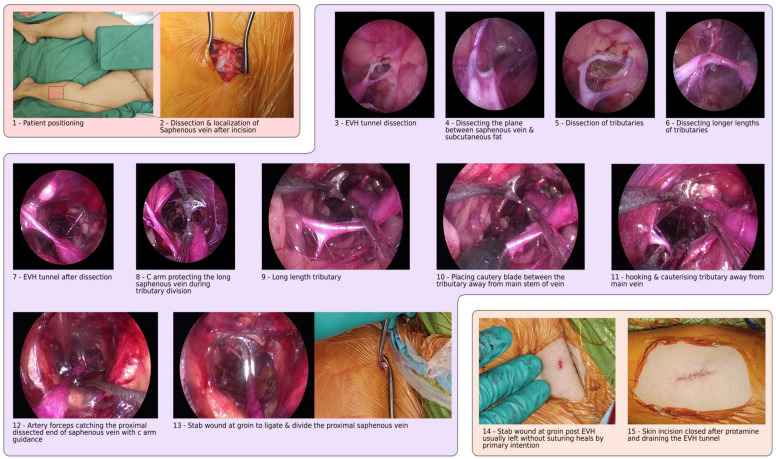
Step-by-step instructions for using the endoscopic vein-harvesting procedure, from patient positioning to wound closure.

**Figure 2 jcm-13-03405-f002:**
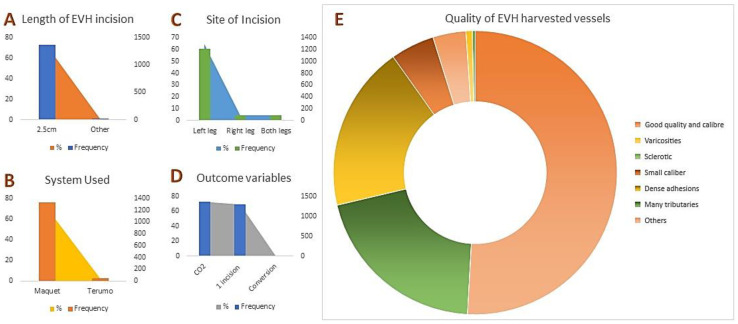
Clinical outcomes of EVH. (**A**) The usual incision length was 2.5 cm (72.2%). (**B**) The Vasoview^TM^ 7xB EVH system was the most frequently used device. (**C**) Left leg incisions were the most common (64.5%). (**D**) The conversion from EVH to OVH occurred in 0.5% of cases. (**E**) Harvested vessel quality varied: 63.8% were of good quality, while 23.7% had dense adhesions. Other categories included varicosities (0.9%), sclerosis (0.4%), small calibers (6.3%), and many tributaries (25.6%).

**Table 1 jcm-13-03405-t001:** Baseline preoperative patient characteristics.

Variable	EVH (*n* = 1418)	OVH (*n* = 466)	Total (*n* = 1884)	*p*-Value
Age, years, mean ± SD	62.1 ± 8.6	61.7 ± 9.3	62.1 ± 8.6	0.436
Race *				0.502
Chinese (%)	935 (65.9)	293 (62.9)	1228 (65.2)	
Malay (%)	276 (19.5)	100 (21.5)	376 (20.0)	
Indian (%)	140 (9.9)	56 (12.0)	196 (10.4)	
Others (%)	67 (4.7)	17 (3.6)	84 (4.5)	
Gender, female (%)	209 (14.7)	74 (15.9)	283 (15.0)	0.550
Diabetes management				0.152
Non-diabetic (%)	652 (46.0)	238 (51.1)	890 (47.2)	
On diet (%)	55 (3.9)	22 (4.7)	77 (4.1)	
Oral therapy (%)	495 (34.9)	147 (31.5)	642 (34.1)	
On insulin (%)	216 (15.2)	59 (12.7)	275 (14.6)	
HbA1c ^$^, mean ± SD	7.7 ± 1.8	7.6 ± 1.9	7.6 ± 1.8	0.103
Hyperlipidemia (%)	1200 (84.6)	409 (87.8)	1609 (85.4)	
Hypertension (%)	1041 (73.4)	348 (74.7)	1389 (73.7)	
Smoking history *				0.918
Non-smoker	648 (46.1)	213 (46.3)	861 (46.1)	
Current smoker	389 (27.6)	127 (27.6)	516 (27.6)	
Ex-smoker	370 (26.3)	120 (26.1)	490 (26.2)	
Smoker pack years ^$^, IQR	30 [15–41]	30 [20–47.5]	30 [16–41.3]	0.173
Peripheral vascular disease ^#^ (%)	135 (9.5)	43 (9.2)	178 (9.4)	0.467
Pulmonary disease (%)	63 (4.4)	20 (4.3)	83 (4.4)	0.505
Carotid disease ^#^ (%)	129 (9.1)	48 (10.3)	177 (9.4)	0.246
Cerebrovascular disease ^#^ (%)	169 (11.9)	52 (11.2)	221 (11.7)	0.363
Left main stem disease ^#^ (%)	442 (31.2)	169 (36.3)	611 (32.4)	0.024 ^S^
Ejection fraction * (EF%)				<0.001 ^S^
Good EF (>49%) (%)	870 (61.4)	244 (52.4)	1114 (59.2)	
Fair EF (30–49%) (%)	409 (28.9)	156 (33.5)	565 (30.0)	
Poor EF (<30%) (%)	137 (9.7)	66 (14.2)	203 (10.8)	
Previous CV intervention ^#^ (%)	2 (0.1)	2 (0.4)	4 (0.2)	0.257
Previous PCI ^#^ (%)	317 (22.4)	114 (24.5)	431 (22.9)	0.190
Renal disease *				0.419
No renal disease (%)	1299 (91.6)	422 (90.6)	1721 (91.3)	
Acute renal failure (%)	13 (0.9)	3 (0.6)	16 (0.8)	
Chronic renal failure (%)	103 (7.3)	40 (8.6)	143 (7.6)	
Functioning transplant (%)	3 (0.2)	1 (0.2)	4 (0.2)	
Creatinine level ^$^ mmol/L, IQR	86 [73–104]	85 [71–106.3]	86 [72–104]	0.586
Operative urgency				<0.001 ^S^
Elective (%)	875 (61.7)	268 (57.5)	1143 (60.7)	
Emergency (%)	10 (0.7)	26 (5.6)	36 (1.9)	
Salvage (%)	0 (0.0)	5 (1.1)	5 (0.3)	
Urgent (%)	533 (37.6)	167 (35.8)	700 (37.2)	
Poor mobility ^#^ (%)	52 (3.7)	11 (2.4)	63 (3.3)	0.110
Logistic EUROscore, IQR	2.5 [1.5–4.6]	2.9 [1.6–7.5]	2.6 [1.5–5.1]	<0.001 ^S^

Data were frequency, *n* (%), analyzed by the Chi-square test. ^S^ = significant; * = linear-by-linear association; ^#^ = Fisher’s exact test; ^$^ = Mann–Whitney U; SD = standard deviation; IQR = interquartile range; EF = ejection fraction; CV = cardiovascular; and PCI = percutaneous coronary intervention.

**Table 2 jcm-13-03405-t002:** Postoperative clinical outcomes, including leg wound complications and other complications.

Outcome Variable	EVH (*n* = 1418)	OVH (*n* = 466)	Total (*n* = 1884)	*p*-Value
Number of arterial grafts (%)				<0.001 ^S^
0	29 (2.0)	33 (7.1)	62 (3.3)	
1	1275 (89.9)	391 (83.9)	1666 (88.4)	
2	102 (7.2)	36 (7.7)	138 (7.3)	
3	12 (0.8)	5 (1.1)	17 (0.9)	
4	0 (0.0)	1 (0.2)	1 (0.1)	
Number of venous grafts (%)				<0.001 ^S^
0	24 (1.7)	36 (7.7)	60 (3.2)	
1	275 (19.4)	82 (17.6)	357 (18.9)	
2	777 (54.8)	240 (51.5)	1017 (54.0)	
3	315 (22.2)	94 (20.2)	409 (21.7)	
4	26 (1.8)	13 (2.8)	39 (2.1)	
5	1 (0.1)	1 (0.2)	2 (0.1)	
Length of the procedure (min), IQR	285 [251.8–319]	273 [242–310]	281 [249–317]	<0.001 ^S^
Cumulative bypass time (min), IQR	129 [103.5–159]	126 [100–154]	128 [102–157]	0.020 ^S^
Cumulative cross-clamp time (min), IQR	72 [55–87.3]	68 [49–82]	71 [53–86]	<0.001 ^S^
Leg wound complication ^#^ (%)	264 (18.6)	149 (32.0)	413 (21.9)	<0.001 ^S^
Leg wound infection ^#^ (%)	75 (5.3)	38 (8.2)	113 (6.0)	0.032 ^S^
Leg wound purulent discharge ^#^ (%)	10 (0.7)	3 (0.6)	13 (0.7)	0.593
Leg wound pain ^#^ (%)	24 (1.7)	11 (2.4)	35 (1.9)	0.330
Leg wound hematoma ^#^ (%)	46 (3.2)	6 (1.3)	52 (2.8)	0.023 ^S^
Leg wound cellulitis^#^ (%)	25 (1.8)	14 (3.0)	39 (2.1)	0.131
Leg oedema (%)	46 (3.3)	16 (3.6)	62 (3.4)	0.589
Reop due to bleeding/tamponade (%)	30 (2.1)	24 (5.2)	54 (2.9)	0.001 ^S^
Renal failure ^#^ (%)	38 (2.7)	12 (2.6)	50 (2.7)	0.528
Stroke (permanent) ^#^ (%)	27 (1.9)	8 (1.7)	35 (1.9)	0.448
Stroke (transient) ^#^ (%)	21 (1.5)	6 (1.3)	27 (1.4)	0.483
Patient status at discharge (%)				0.115
Alive—well	1366 (96.9)	440 (95.0)	1806 (96.4)	
Alive with minor complication	22 (1.6)	9 (1.9)	31 (1.7)	
Dead	22 (1.6)	14 (3.0)	36 (1.9)	
Readmission within 30 days ^#^ (%)	228 (16.1)	78 (16.7)	306 (16.2)	0.772
Operative mortality	22 (1.6)	14 (3.0)	36 (1.9)	0.047 ^S^
Cardiac mortality ^#^ (%)	7 (0.5)	5 (1.1)	12 (0.6)	0.173
MACEs (composite)	55 (3.9)	18 (3.9)	73 (3.9)	0.988

Data were frequency, *n* (%), analyzed by the Chi-square test. *p* < 0.05 is statistically significant. ^S^ = significant; ^#^ = Fisher’s exact test; IQR = interquartile range; and MACE = major adverse cardiovascular events.

## Data Availability

The data predominantly appear in Section 3 and Appendix A. Additional data are available upon request.

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
