# Peer review of "A Comprehensive Clinical Outcome Analysis of Endoscopic Vessel Harvesting for Coronary Artery Bypass Surgery"

_jcm, 2024, doi:10.3390/jcm13123405_

Round 1
Reviewer 1 Report
Comments and Suggestions for Authors
The introduction provides a good overview of the background and significance of the study. However, it could benefit from a more detailed explanation of why EVH is considered over OVH in certain clinical scenarios. Including a brief discussion on the current guidelines or standard practices would strengthen the context.
The methods section is comprehensive but could benefit from more detail on the statistical methods used, particularly the logistic regression models. Providing a rationale for selecting specific covariates in the multivariate analysis would enhance clarity. The study mentions the use of specific devices for EVH. It would be helpful to include a brief description or reference for these devices for readers unfamiliar with them.
The results are clearly presented with appropriate use of tables and figures. However, the narrative could be streamlined to avoid repetition. For instance, the differences in baseline characteristics could be summarized more succinctly. The p-values are appropriately reported, but the clinical significance of the findings could be emphasized more. Discussing the absolute differences and their potential impact on clinical practice would be beneficial.
The discussion is well-structured and addresses the main findings. However, the limitations section should be expanded to include potential biases due to the retrospective nature of the study and the single-center design. Comparing the findings more extensively with previous studies would provide a deeper context. Highlighting similarities and differences with other research can help in understanding the broader implications. The authors mention the need for specialized training for EVH. It would be useful to discuss any specific training protocols or recommendations for institutions looking to adopt EVH.
The conclusion effectively summarizes the main findings but should reiterate the potential clinical implications and recommendations for future research more strongly. Suggesting specific areas for further investigation, such as long-term outcomes or cost-effectiveness, would be helpful.
Figures and tables are informative but could be improved by ensuring consistency in labeling and presentation. For instance, ensuring all abbreviations are defined in the table legends and maintaining uniform font sizes would enhance readability.
The reference list is comprehensive but should ensure the most recent and relevant studies are included. Given the rapid advancements in surgical techniques, ensuring up-to-date references will strengthen the manuscript.
Author Response
Dear Editor and Reviewers
Thank you to all the reviewers for investing their time into reading our manuscript. We have made most of the requested changes to the manuscript and summarised all our edits below. All changes made to the manuscript have been highlighted with “Yellow” to facilitate ease of further review. The reviewers' comments have been bolded in this document, and responses are in “Blue.” Once again, a wholehearted thank you to all our reviewers.
Reviewer 1
Comments and Suggestions for Authors
The introduction provides a good overview of the background and significance of the study. However, it could benefit from a more detailed explanation of why EVH is considered over OVH in certain clinical scenarios. Including a brief discussion on the current guidelines or standard practices would strengthen the context.
Response:
Thank you for your valuable feedback. An explanation of why EVH is considered over OVH is added (Lines 39-42):
“EVH is associated with a significantly lower incidence of postoperative complications such as leg wound infections and other wound complications like hematomas, seromas, and dehiscence. These reduced complications contribute to better overall patient outcomes.”
A brief discussion on the current guidelines or standard practices is added (Lines 45-48):
“Patients undergoing EVH typically experience less postoperative pain, quicker recovery times, and faster mobilization, which are crucial for reducing hospital stays and overall healthcare costs. Current guidelines from the Society of Thoracic Surgeons (STS) and the European Association for Cardio-Thoracic Surgery (EACTS) support the use of EVH due to these benefits.”
Certain clinical scenarios have been described (Lines 49-55):
“EVH is thus often favored over OVH in certain clinical scenarios due to its numerous advantages. EVH is particularly advantageous for high-risk patients, such as those with diabetes, obesity, or peripheral vascular disease, as well as for elderly patients and those requiring multiple grafts. The reduced morbidity and faster recovery associated with EVH make it a preferred method in many hospitals and surgical centers, which have incorporated it into their standard practices and training programs for cardiovascular surgeons.”
The methods section is comprehensive but could benefit from more detail on the statistical methods used, particularly the logistic regression models. Providing a rationale for selecting specific covariates in the multivariate analysis would enhance clarity. The study mentions the use of specific devices for EVH. It would be helpful to include a brief description or reference for these devices for readers unfamiliar with them.
Response:
Thank you for your valuable comments.
We have incorporated a sub-section under methods to describe the EVH devices. Subsection 2.4 (Lines 132-146):
“The Vasoview™ 7xB EVH System by Maquet, Getinge Group, and the VirtuoSaph™ Plus EVH System by Terumo Cardiovascular offer advanced solutions for endoscopic vessel harvesting, each with distinct features and functionalities. While both systems aim to optimize endoscopic vessel harvesting, they differ in their design nuances, such as dissection techniques and device ergonomics, catering to varied surgeon preferences and clinical settings. The VasoviewTM 7xB EVH system (Maquet, Getinge Group, Karlsruhe, Germany) provides high-definition endoscopic visualization and employs a blunt dis-section technique, minimizing tissue trauma, while also incorporating bipolar electro-cautery for efficient hemostasis. Its ergonomic design enhances surgeon comfort and control, with the added benefit of CO2 insufflation for a clear operative field (Figure 1). VirtuoSaphTM Plus EVH System (Terumo Cardiovascular, Ann Arbor, MI, USA) similarly prioritizes precise visualization and utilizes a semi-blunt, atraumatic dissection approach to preserve vein integrity. It features an intuitive control interface and single-use com-ponents for sterility maintenance, alongside a disposable bipolar cautery system for hemostasis.”
Statistical methods and rationale for selecting specific covariates have been elaborated (Lines 158-166):
“Univariate and multivariate logistic regression analyses were used to determine the as-sociation between vessel harvesting methods (EVH vs. OVH) and leg wound complications. By adjusting for confounders and prognostic factors, these analyses provide an accurate estimate of the harvesting method's independent effect. This approach yields adjusted odds ratios, indicating the strength and significance of the association, ensuring reliable results. Logistic regression was used to compare operational outcomes between the two harvesting methods to adjust for important prognostic factors or confounders detected using bivariate analysis.”
The results are clearly presented with appropriate use of tables and figures. However, the narrative could be streamlined to avoid repetition. For instance, the differences in baseline characteristics could be summarized more succinctly. The p-values are appropriately reported, but the clinical significance of the findings could be emphasized more. Discussing the absolute differences and their potential impact on clinical practice would be beneficial.
Response:
We have concise the result section and removed the repetition of the data that has been included in Table 1.
First paragraph of result section was “The study cohort comprised a total of 1935 patients. As some patients were lost to follow-up because of various factors, including loss of data, it is crucial to note that only 1884 patients were subjected to analysis of postoperative complications in the short-term and intermediate follow-up period, and 51 patients (2.6%) were excluded from the final analysis; these patients were treated with a combination of EVH and OVH or underwent conversion. Among these patients, 75.3% were in the EVH group, and 24.7% were in the OVH group. The sample had 14.7% females, and 65.9% were of Chinese ethnicity.”
This section has been modified to (Lines 168-170):
“Of the initial 1935 patients, 1884 were analyzed for postoperative complications, with 51 excluded due to mixed or converted procedures. 75.3% underwent EVH, while 24.7% underwent OVH. The sample included 14.7% females, with 65.9% of Chinese ethnicity.”
Sub-section 3.1 was “Before we analyzed the postoperative complications in our study population, we conducted an analysis to determine the preoperative baseline characteristics of the study group. Table 1 shows the important baseline characteristics that were clinically relevant to our study. The analysis was imperative to ensure that there was no significant differ-ence in preoperative characteristics between the EVH and OVH groups. However, the ejection fraction (EF), operative urgency, and logistic EUROscore were significantly dif-ferent between the groups. More patients had a Fair EF (30-49%) or a poor EF (<30%) in the OVH group, and more patients had a Good EF (>49%) in the EVH group (p < 0.001). Operative urgency was clinically significant in the EVH group (p < 0.001), and the Lo-gistic EUROscore was higher in the OVH group. Race showed no clinical significance (p = 0.100). Other baseline characteristics were clinically insignificant in group comparisons.”
This sub-section has been modified to (Lines 176-180):
“Before analyzing postoperative complications, we assessed the preoperative base-line characteristics of our study population. Table 1 summarizes clinically relevant factors. This analysis aimed to ensure comparability between the EVH and OVH groups. Significant differences were observed in ejection fraction (EF), operative urgency, and logistic EUROscore.”
Similarly, sub-section 3.2 was “With respect to the overall clinical outcomes of 1884 CABG patients, we found a low incidence of renal complications, with 2.2% experiencing acute kidney injury (AKI) and 0.6% suffering renal failure. Cardiac-related deaths accounted for 0.6% (12 patients), while perioperative myocardial infarctions were absent. Saphenous vein usage prevailed (73.6%), with 4.5% utilizing the radial artery. Most patients had 3 distal coronary anas-tomoses (56.8%) and underwent cardiopulmonary bypass (97.1%), primarily through aortic and atrial/caval cannulation. The VasoviewTM 7xB EVH system (Maquet, Getinge Group, Karlsruhe, Germany) was predominantly employed for EVH, with left leg inci-sions being most common (64.5%). The other device used for the EVH procedures was the VirtuoSaphTM Plus EVH System from Terumo Cardiovascular, Ann Arbor, MI, USA, used in 2.8% of procedures. Most EVH incisions were 2.5 cm, and vessel quality varied, with 63.8% deemed good quality (Figure 2). Further details are available in Supplemen-tary Tables 1,2.”
This section has been modified to (Lines 178-188):
“In the overall clinical outcomes of 1884 CABG patients, renal complications were low, with 2.2% experiencing acute kidney injury (AKI) and 0.6% suffering renal failure. Cardiac-related deaths accounted for 0.6% (12 patients), with no perioperative myocardial infarctions reported. Saphenous vein usage prevailed (73.6%), with 4.5% utilizing the radial artery. Most patients had 3 distal coronary anastomoses (56.8%) and underwent cardiopulmonary bypass (97.1%), primarily through aortic and atrial/caval cannulation. The Vasoview™ 7xB EVH system was predominantly used for EVH, with left leg incisions being most common (64.5%). The VirtuoSaph™ Plus EVH System from Terumo Cardiovascular was used in 2.8% of procedures. Most EVH incisions were 2.5 cm, with 63.8% of vessels deemed of good quality (Figure 2). Further details can be found in Supplementary Tables 1, 2.”
The absolute differences between EVH and OVH have been highlighted in sub-section 3.3. This section has been modifies to (Lines 190-197):
“Table 2 presents bivariate comparisons of clinical outcomes between the study groups. EVH was associated with a significantly higher number of arterial and venous grafts, longer procedure, cumulative bypass time, and cumulative cross-clamp times (p<0.001 for all). Postoperative leg wound infections were less frequent with EVH (5.3% vs. 8.2%, p<0.001), while EVH showed a higher incidence of leg wound hematoma com-pared to OVH (3.2% vs. 1.3%, p<0.001). No significant differences were observed in other complications, renal failure, stroke, discharge status, readmission within 30 days, cardiac mortality, or major adverse cardiovascular events (MACE).”
The discussion is well-structured and addresses the main findings. However, the limitations section should be expanded to include potential biases due to the retrospective nature of the study and the single-center design. Comparing the findings more extensively with previous studies would provide a deeper context. Highlighting similarities and differences with other research can help in understanding the broader implications. The authors mention the need for specialized training for EVH. It would be useful to discuss any specific training protocols or recommendations for institutions looking to adopt EVH.
Response:
Thank you for your valuable feedback. We have addressed your suggestion and made changes in a point-by-point manner as listed below:
- Comparing the findings more extensively with previous studies (Lines 232-235):
“In a meta-analysis, Deppe et al. reported that EVH is safe for patients under-going CABG, as it reduces leg wound infections. They analyzed a total of 27,789 patients from 43 studies (16 RCTs and 27 observational trials) who underwent saphenectomy: 46% EVH and 54% OVH [16].”
Also in Lines 247-251
“Furthermore, a large retrospective multicenter analysis in-dicates that EVH has a lower risk of mortality compared to OVH over a follow-up period of six years. Although our study did not include an extended follow-up period, similar findings are expected and have been reported in other studies [22].”
- We have included a brief description of the need for specialized training (Lines 243-246):
“Additionally, the learning curve associated with EVH, particularly with devices such as VasoviewTM (Marquet) and VirtuoSaphTM (Terumo), could prolong surgeon adaptation, causing more reluctance to perform the operation. Hematoma may be more prevalent during this acclimation period, underscoring the complex interplay of factors affecting postprocedural outcomes [21].”
However, we feel, a detailed discussion regarding the EVH training is beyond the scope of this manuscript. Hence further edition in this section was not made.
- Limitation section has been modified based on your comments. We have included statement regarding potential biases due to the retrospective nature of the study and the single-center design (Lines 262-272):
“This study has a few limitations. These include potential biases due to its retrospective nature and single-center design, which precluded randomization or blinding, potentially introducing bias between the two groups. The absence of randomization might have contributed to baseline disparities that could influence the study outcomes. Nevertheless, there were no substantial differences in baseline characteristics between the EVH and OVH groups, except in cases of operative urgency. However, the groups did differ in significant clinical indicators such as Logistic EUROscore, ejection fraction, and left main stem disease, which may have affected the clinical results. Notably, patients requiring urgent operations preferred OVH over EVH, likely because OVH is a faster procedure. Additionally, since the investigation was conducted in a single center, the patient population might not be representative of the general population.”
The conclusion effectively summarizes the main findings but should reiterate the potential clinical implications and recommendations for future research more strongly. Suggesting specific areas for further investigation, such as long-term outcomes or cost-effectiveness, would be helpful.
Response:
Thank you for your comment. We have modified the conclusion based on your suggestion (Lines 274-280):
“Our findings suggest that EVH reduces the incidence of postoperative complications, providing benefits such as a lower risk of complications, improved cosmetic outcomes, and faster recovery in CABG surgery. However, the consideration of operative urgency and the necessity for specialized training in EVH should inform its implementation. To solidify these findings, further prospective studies are crucial. Future research should focus on long-term outcomes, cost-effectiveness, and identifying the optimal conditions for EVH to enhance care quality for CABG patients.”
Figures and tables are informative but could be improved by ensuring consistency in labeling and presentation. For instance, ensuring all abbreviations are defined in the table legends and maintaining uniform font sizes would enhance readability.
Response:
Thank you for your comment. We have changed the tile and legends of all tables and figurers as per your suggestion. The font size was also converted to maintain uniform nature.
The reference list is comprehensive but should ensure the most recent and relevant studies are included. Given the rapid advancements in surgical techniques, ensuring up-to-date references will strengthen the manuscript.
Response:
Thank you for your comment. Two recent references are included.
Reviewer 2 Report
Comments and Suggestions for Authors
I am grateful to the editor for the opportunity to review the manuscript of Hari Kumar Sampath et al, “A Comprehensive Clinical Outcome Analysis of Endoscopic Vessel Harvesting for Coronary Artery Bypass Surgery.” In this manuscript, the authors presented the clinical results of endoscopic vascular harvesting for coronary artery bypass grafting in comparison with open harvesting of the great saphenous vein. This issue is the subject of debate among coronary surgeons. It is clear that endoscopic vein harvesting should cause fewer complications from the site of its collection than the open method. Whether this method affects the long-term patency of shunts remains the subject of research. This publication cannot answer the last question, since it evaluates only the immediate clinical results of CABG.
While reviewing, I had the following questions that I would like answers to from the authors:
1. There is no purpose of the study in the text of the manuscript; it is present only in the ABSTRACT section.
2. Since there are many publications on this issue, including meta-analyses (ref. 11-12 in the article, as well as ref. 1, see below), it is necessary to substantiate the novelty of this study in more detail. To do this, you can use data from other studies with a larger number of patients included (for example, ref. 2 cm below)
3. In section 4.1. Limitations authors state: "... there were no substantial differences in the baseline characteristics between the EVH and OVH groups, except in cases of operative urgency" (lines 231-232). However, the groups differed in a number of significant clinical indicators (Logistic EUROscore, Ejection fraction, Left main stem disease). This may have affected the clinical results, so these facts should also be noted in the Limitations of the study section.
4. In section 5. Conclusions, the authors state: "Our findings suggest that EVH reduces the incidence of postoperative complications, offering advantages like reduced risk of complications, ..." (lines 237-238). However, for such indicators as "Reop due to bleeding/tamponade" and "Operative mortality" the OVH group performed worse than the EVH group. This could be due to differences in clinical status (see above). This should have been addressed in the Discussion section and the wording of the conclusion adjusted accordingly.
References:
1. Deppe AC, Liakopoulos OJ, Choi YH, Slottosch I, Kuhn EW, Scherner M, Stange S, Wahlers T. Endoscopic vein harvesting for coronary artery bypass grafting: a systematic review with meta-analysis of 27,789 patients. J Surg Res. 2013 Mar;180(1):114-24. doi: 10.1016/j.jss.2012.11.013.
2. Krishnamoorthy B, Zacharias J, Critchley WR, Rochon M, Stalpinskaya I, Rajai A, Venkateswaran RV, Raja SG, Bahrami T. A multicentre review comparing long-term outcomes of endoscopic vein harvesting versus open vein harvesting for coronary artery bypass surgery. NIHR Open Res. 2021 Jul 8;1:11. doi: 10.3310/nihropenres.13215.1.
Comments on the Quality of English LanguageNo comments
Author Response
Dear Editor and Reviewers
Thank you to all the reviewers for investing their time into reading our manuscript. We have made most of the requested changes to the manuscript and summarised all our edits below. All changes made to the manuscript have been highlighted with “Yellow” to facilitate ease of further review. The reviewers' comments have been bolded here, and responses are in “Blue.” Once again, a wholehearted thank you to all our reviewers.
Reviewer 2
Comments and Suggestions for Authors
I am grateful to the editor for the opportunity to review the manuscript of Hari Kumar Sampath et al, “A Comprehensive Clinical Outcome Analysis of Endoscopic Vessel Harvesting for Coronary Artery Bypass Surgery.” In this manuscript, the authors presented the clinical results of endoscopic vascular harvesting for coronary artery bypass grafting in comparison with open harvesting of the great saphenous vein. This issue is the subject of debate among coronary surgeons. It is clear that endoscopic vein harvesting should cause fewer complications from the site of its collection than the open method. Whether this method affects the long-term patency of shunts remains the subject of research. This publication cannot answer the last question, since it evaluates only the immediate clinical results of CABG.
Response:
Thank you for your positive comments.
While reviewing, I had the following questions that I would like answers to from the authors:
- There is no purpose of the study in the text of the manuscript; it is present only in the ABSTRACT section.
Response:
Thank you for your comment. We have included the purpose of the study in the abstract and introduction section.
The introduction shows (Lines 64-68): “Notwithstanding these limitations, ongoing research and technological advancements are likely to address these issues, enhancing the overall effectiveness and applicability of EVH in cardiovascular surgery. The aim of our study was to directly compare the clinical outcomes of EVH with those of OVH to uncover potential disparities in clinical application.”
- Since there are many publications on this issue, including meta-analyses (ref. 11-12 in the article, as well as ref. 1, see below), it is necessary to substantiate the novelty of this study in more detail. To do this, you can use data from other studies with a larger number of patients included (for example, ref. 2 cm below)
Response:
Thank you for your valuable feedback. The following section with suggested references have been added to the manuscript:
Lines 232-235:
“In a meta-analysis, Deppe et al. reported that EVH is safe for patients under-going CABG, as it reduces leg wound infections. They analyzed a total of 27,789 patients from 43 studies (16 RCTs and 27 observational trials) who underwent saphenectomy: 46% EVH and 54% OVH [16].”
Also in Lines 247-251
“Furthermore, a large retrospective multicenter analysis in-dicates that EVH has a lower risk of mortality compared to OVH over a follow-up period of six years. Although our study did not include an extended follow-up period, similar findings are expected and have been reported in other studies [22].”
- In section 4.1. Limitations authors state: "... there were no substantial differences in the baseline characteristics between the EVH and OVH groups, except in cases of operative urgency" (lines 231-232). However, the groups differed in a number of significant clinical indicators (Logistic EUROscore, Ejection fraction, Left main stem disease). This may have affected the clinical results, so these facts should also be noted in the Limitations of the study section.
Response:
Thank you for your valuable feedback. Limitation section has been modified based on your comments (Lines 262-272):
“This study has a few limitations. These include potential biases due to its retrospective nature and single-center design, which precluded randomization or blinding, potentially introducing bias between the two groups. The absence of randomization might have contributed to baseline disparities that could influence the study outcomes. Nevertheless, there were no substantial differences in baseline characteristics between the EVH and OVH groups, except in cases of operative urgency. However, the groups did differ in significant clinical indicators such as Logistic EUROscore, ejection fraction, and left main stem disease, which may have affected the clinical results. Notably, patients requiring urgent operations preferred OVH over EVH, likely because OVH is a faster procedure. Additionally, since the investigation was conducted in a single center, the patient population might not be representative of the general population.”
- In section 5. Conclusions, the authors state: "Our findings suggest that EVH reduces the incidence of postoperative complications, offering advantages like reduced risk of complications, ..." (lines 237-238). However, for such indicators as "Reop due to bleeding/tamponade" and "Operative mortality" the OVH group performed worse than the EVH group. This could be due to differences in clinical status (see above). This should have been addressed in the Discussion section and the wording of the conclusion adjusted accordingly.
Response:
Thank you for your comments. We have modified the discussion section as above. And, we have also modified the conclusion based on your suggestion (Lines 274-280):
“Our findings suggest that EVH reduces the incidence of postoperative complications, providing benefits such as a lower risk of complications, improved cosmetic outcomes, and faster recovery in CABG surgery. However, the consideration of operative urgency and the necessity for specialized training in EVH should inform its implementation. To solidify these findings, further prospective studies are crucial. Future research should focus on long-term outcomes, cost-effectiveness, and identifying the optimal conditions for EVH to enhance care quality for CABG patients.”
References:
- Deppe AC, Liakopoulos OJ, Choi YH, Slottosch I, Kuhn EW, Scherner M, Stange S, Wahlers T. Endoscopic vein harvesting for coronary artery bypass grafting: a systematic review with meta-analysis of 27,789 patients. J Surg Res. 2013 Mar;180(1):114-24. doi: 10.1016/j.jss.2012.11.013.
- Krishnamoorthy B, Zacharias J, Critchley WR, Rochon M, Stalpinskaya I, Rajai A, Venkateswaran RV, Raja SG, Bahrami T. A multicentre review comparing long-term outcomes of endoscopic vein harvesting versus open vein harvesting for coronary artery bypass surgery. NIHR Open Res. 2021 Jul 8;1:11. doi: 10.3310/nihropenres.13215.1.
Response:
Thank you for your comment. The references are included.
Comments on the Quality of English Language: No comments
Response:
Thank you for your feedback.
Round 2
Reviewer 2 Report
Comments and Suggestions for Authors
The authors responded to my comments and made corrections to the text. I have no other comments.
Comments on the Quality of English LanguageNo comments